# Impact of dietary aflatoxin on immune development in Gambian infants: a cohort study

Ya Xu,[1] Sophie Moore,[2] Gaoyun Chen,[3] Patrick Nshe,[4] Tisbeh Faye-Joof,[4] Andrew Prentice,[4] Yun Yun Gong,[5] Michael Routledge [1,6]

[1]School of Medicine, University of Leeds, Leeds, UK
[2]Department of Women and Children's Health, King's College London, London, UK
[3]Queen's University Belfast, Belfast, UK
[4]MRC Unit The Gambia at LSHTM, Banjul, Gambia
[5]School of Food Science and Nutrition, University of Leeds, Leeds, UK
[6]School of Food and Biological Engineering, Jiangsu University, Zhenjiang, China

**Correspondence to**
Dr Michael Routledge;
medmnr@leeds.ac.uk

## ABSTRACT

**Background** Chronic aflatoxin (AF) exposure has been shown to occur at high levels in children from sub-Saharan Africa (SSA), and has been associated with growth retardation and immune dysfunction. Our objective was to investigate the impact of AF exposure on immune development in early infancy using thymic size and antibody (Ab) response to vaccination as indicators of immune function.

**Methods** A total of 374 infants born between May 2011 and December 2012 were enrolled into the current study. These infants were recruited from a larger, randomised trial examining the impact of nutritional supplementation of mothers and infants on infant immune development (the Early Nutrition and Immune Development Trial). Thymic size (Thymic Index, TI) was measured by sonography at 1 week, 8 weeks, 24 weeks and 52 weeks of infant age. Infants were given the diphtheria–tetanus–pertussis (DTP) vaccine at 8 weeks, 12 weeks and 16 weeks of age, and Ab responses to each vaccine measured at 12 weeks and 24 weeks of age. AF-albumin (AF-alb) adduct levels in infant blood were measured by ELISA as the biomarker of AF exposure.

**Results** The geometric mean (GM) level of AF-alb increased with age. Only half of infants had detectable AF-alb with a GM of 3.52 pg/mg at 24 weeks, increasing to 25.39 pg/mg at 52 weeks, when 98% of infants had AF-alb >limit of detection. Significant negative association of AF-alb level with TI was seen in infants during the first 24 weeks, especially at 8 weeks of age (p<0.001), which is the time point of fastest thymus growth. There were no associations between AF exposure level and Ab response to pertussis and tetanus, but a significant positive correlation was observed between AF-alb level and Ab titre to diphtheria (p<0.005).

**Conclusions** High levels of AF exposure during early infancy may impact on infant immune development.

**Trial registration number** ISRCTN49285450.

## INTRODUCTION

Aflatoxin (AF) is a mycotoxin produced mainly by *Aspergillus flavus* and *Aspergillus parasiticus* fungi that frequently contaminate crops in tropical and subtropical areas. It poses a great health risk to populations living in sub-Saharan Africa (SSA), especially in regions where groundnuts and/or maize are

### Strengths and limitations of this study

► The study was embedded within a clinical trial (Early Nutrition and Immune Development), which reduced selection bias of subjects.
► The subjects were part of a cohort naturally exposed to aflatoxin (AF) through their diet, which allowed for multiple time point sampling.
► The AF exposure was measured using a validated biomarker, which allows better estimation of exposure compared with food intake estimates.
► A limitation of the study is that AF exposure in utero (ie, by measuring biomarker levels in the pregnant women) was not performed.

the staple foods.[1] Exposure to AF can begin in utero through transplacental exposure, continue in early infancy through breast feeding and increase as children are weaned onto family foods, such as maize porridge or peanut sauces.[2] Populations in the Gambia are at a high risk of AF exposure through food consumption.[3–7]

AF is a human liver carcinogen,[8] and has also been associated with childhood growth faltering.[9 10] An inverse relationship has been reported between AF-albumin adduct (AF-alb) levels in pregnant women and subsequent infant growth[5] as well as between AF-alb and growth in infants below 2 years old,[7 10] and high levels of AF lysine were associated with stunting and severe malnutrition in children in Nigeria,[11] and with underweight children in Kenya.[12] Data from animal studies suggests that AF modulates the immune response at the level of innate cell functions, antibody (Ab) production, lymphocyte activation and proliferation and regulation of cytokine/chemokine expression, but there have been few studies on the impact of AF on immune function in humans.[13 14] A reduction of salivary IgA expression[15] and lower percentage of lymphocytes[16 17] in children with high AF exposure have been reported. A greater

understanding of the impact of chronic AF exposure during infancy and early childhood on immune system development is required.

The thymus is a primary lymphoid organ, essential for the development and differentiation of T lymphocytes. Thymic Index (TI; a derived estimate of thymic volume) measured sonographically has been used as an indicator of immune development in infants. Impaired thymic development in infancy has been associated with morbidity and mortality in childhood in Guinea-Bissau and Bangladesh,[18–20] indicating a functional consequence of poor early thymic development.

The effect of chronic AF exposure on humoral immunity is less consistent in animal and human studies.[21 22] A study conducted in infants in the Gambia reported a weak but significant positive association between the level of AF-alb and pneumococcal Ab titres.[15] Combined DTP (diphtheria, tetanus and pertussis) vaccine has been involved in the Expanded Programme on Immunisation (EPI), which was established in 1977 by the WHO and is estimated to prevent 2–3 million deaths in children every year.

The aim of the current study was to determine the impact of AF exposure in early infancy on thymus growth and altered Ab response to combined DTP vaccination.

## METHODS
### Study subjects
The current substudy into the effects of AF exposure on immune parameters was embedded within the Early Nutrition and Immune Development (ENID) trial. Full details of the ENID trial protocol have been published.[23] The primary objective of the ENID trial was to investigate whether combined prenatal (protein energy and/or micronutrients) and infant (micronutrient) nutritional supplements could improve child's immune development. In total, 875 pregnant women from the West Kiang region of the Gambia were recruited into the ENID trial. In the current study, blood samples were taken from all children born into the ENID trial between May 2011 and December 2012 (N=374), to give sufficient power to assess impact of AF exposure on TI and Ab response.

At enrolment into the ENID trial, pregnant women were randomly allocated to four different trial arms: iron–folate (FeFol), multiple micronutrients (MMN), protein energy (PE) combined with FeFol and PE combined with MMN. Supplementation continued until delivery. From 6 months to 12 months of age, infants were further randomised into two supplementation arms: lipid-based nutritional supplementation (LNS) or LNS+MMN. Compliance of the supplementation was also recorded, through weekly interview. Due to the rolling randomisation into supplementation groups on enrolment, the children from whom samples were used in the current study were randomised into the various groups at approximately equal numbers (table 1).

**Table 1** Characteristics of children in the current study

| Variable | n | Mean±SD/n (%) |
|---|---|---|
| Gender, n (%) | 374 | |
| Male | | 192 (51.3) |
| Female | | 182 (48.7) |
| Ethnicity, n (%) | 348 | |
| Fula | | 30 (8.6) |
| Jola | | 11 (3.2) |
| Mandinka | | 304 (87.4) |
| Other | | 3 (0.9) |
| Mother's education, n (%) | 352 | |
| <1 year formal education | | 235 (66.8) |
| >1 year formal education | | 117 (33.2) |
| Birth weight (kg) | 335 | 3.0±0.4 |
| Birth length (cm) | 335 | 49.7±2.0 |
| Age of introduction of non-breast milk foods (weeks) | | |
| <6 months | 248 | 4.6±1.3 |
| >6 months | 125 | 6.2±0.1 |
| Morbidity | | |
| Sum of first 12 weeks of life | 360 | 8.3±11.0 |
| Sum of first 24 weeks of life | 361 | 18.7±18.3 |
| Maternal supplementation group, n (%) | 374 | |
| FeFol | | 95 (25.4) |
| MMN | | 96 (25.7) |
| PE+FeFol | | 89 (23.7) |
| PE+MMN | | 94 (25.1) |
| Infant supplementation group (6 months), n (%) | 374 | |
| LNS+MMN | | 192 (51.3) |
| LNS only | | 182 (48.7) |
| **AF-alb (pg/mg)** | | **GM (95% CI)** |
| Week 24 | 352 | 3.52 (3.15 to 3.94) |
| Week 52 | 331 | 25.39 (22.37 to 28.82) |

AF-alb, alfatoxin–albumin; FeFol, iron–folate; GM, geometric mean; LNS, lipid-based nutritional supplementation; MMN, multiple micronutrients; PE, protein energy.

For the current analysis, anthropometric measurements collected at 1 week, 8 weeks, 24 weeks and 52 weeks of infant age and infant serum samples collected at 12 weeks, 24 weeks and 52 weeks of age were used. Infant morbidity and feeding practices were recorded by field assistants who visited the children weekly at home. Morbidity questionnaires collected specific data on any episodes of diarrhoea, rapid breathing, vomiting, cough or fever or other symptoms during previous 7 days. At the same weekly visit, data on infant feeding practices (breastfeeding practices and introduction of non-breast milk foods) were also collected, using a standardised questionnaire.

## Thymus size assessment

Thymus size of infants was assessed sonographically at 1 week, 8 weeks, 24 weeks and 52 weeks of age using a validated method.[24] The transverse diameter of the thymus and the sagittal area of its largest lobe were detected and multiplied to give a volume-related TI. Full details of the TI measurements of the ENID trial can be found in Moore *et al*.[25]

## Ab response to vaccination assessment

All infants were immunised in accordance with the Gambia EPI programme.[23] Three injections of combined DTP were given to children at week 8, week 12 and week 16. Venous blood samples collected from infants at 12 weeks and 24 weeks and were used to determine the Ab response to vaccination. A multiple immunoassay based on Luminex xMAP technology was used to detect serum-specific IgG Ab responses against diphtheria toxoid, pertussis toxin and tetanus toxin.[26 27] Full details of the outcomes (Ab response to vaccination) of the ENID trial can be found in Okala *et al*.[28]

## AF exposure measurement

AF-alb levels in serum taken at 24 weeks and 52 weeks of age were measured at the University of Leeds using an ELISA method[29] that has been previously validated against dietary intake.[30] In brief, this analysis is performed in four steps: albumin extraction and quantification, hydrolysis of albumin with pronase, purification of AF-alb residues and the competitive ELISA. The ELISA involves the premixing of standards, samples or controls with the rabbit anti-AF-Cl$_2$-BSA polyclonal Ab, followed by competitive ELISA in which remaining unbound antibody can bind to AF ovalbumin conjugates coated on the surface of the ELISA plate well. After washing, the bound primary Ab is detected by incubation with an enzyme-labelled goat anti-rabbit IgG secondary Ab. Control samples at four known concentrations of AF-alb were examined alongside each batch of samples.

Samples were measured at least twice, in triplicate each time, on two different days. Only results with a coefficient of variation less than 25% were accepted. The limit of detection (LOD) of the assay was 3 pg/mg albumin. The AF-alb level in samples less than LOD were assigned a value of 1.5 pg/mg for statistical analyses.

## Statistical analysis

Statistical analysis was performed using STATA V.15 (StataCorp LP). Distribution of AF-alb level, TI and Ab response to vaccination data were skewed and were, therefore, log transformed prior to further analysis. Ab fold change was calculated as the Ab level at 24 weeks divided by Ab level at 12 weeks. Results were divided into low and high AF exposure groups at both 24 weeks and 52 weeks based on the median of AF-alb level at each time point. Student's t-test was used to compare the association of parameters in different seasons or low/high AF exposure level groups.

For other covariates, the season of sampling was categorised as rainy and dry seasons based on the date of TI measurement and blood sample collection for Ab and AF-alb analysis. There is a distinct seasonal pattern in the Gambia with a period of rainfall from July to October (rainy season) and a long dry season between November and June. The age (months) of introduction of non-breast milk foods to infants were recorded. Infant morbidity was coded as sum of the number of morbidity episodes (diarrhoea, vomiting, cough, rapid breathing and fever).

Relationships between AF-alb level and TI or Ab titre were explored using ordinary least squares regression for individual time points, or random effects model for the pooling data of all time points. Data are presented in different models for TI and Ab titres. The correlation between AF-alb level and TI was analysed as model 1: unadjusted; model 2: adjusted for infant size (length), sex and season at TI measurement; model 3: adjusted for infant size (length), season at TI measurement, sex and maternal supplement groups (for age ≤24 weeks) or both maternal and infants supplement groups (for age at 52 weeks). The correlation between AF-alb level and Ab response was analysed as model 1: unadjusted; model 2: adjusted for weight for height z-score (WHZ) and season of sample collection; and model 3: adjusted for WHZ and season of sample collection, sex, maternal supplement groups, haemoglobin (Hb) levels and morbidity.

## Consent to participate

Written informed consent was obtained from all participants.

## Patient and public involvement

Patients or the public were not involved in the design, conduct, reporting or dissemination plans of our research.

## RESULTS

A total of 800 live born infants were delivered into the main ENID trial.[25] For the current substudy, 374 infants were included. Table 1 summarises the characteristics of the participants in the current substudy. More than half of mothers (67%) had less than 1 year of formal education. Twenty five out of 374 (6.7%) children were born with a low birth weight (<2500 g). The mean (SD) duration of exclusive breast feeding was 5.2 months (1.3 months).

## AF exposure

The geometric mean (GM) level of AF-alb was 3.52 pg/mg at 24 weeks of age and increased to 25.39 pg/mg at 52 weeks with 52% and 2% of samples having levels lower than the LOD (3 pg/mg) at the 24 weeks and 52 weeks age point, respectively. Seasonal variations were observed in the exposure (figure 1) with around twofold higher levels of AF-alb determined in samples collected during the dry season compared with the rainy season at both 24 weeks and 52 weeks of age (GM 24 weeks: 2.5 pg/mg

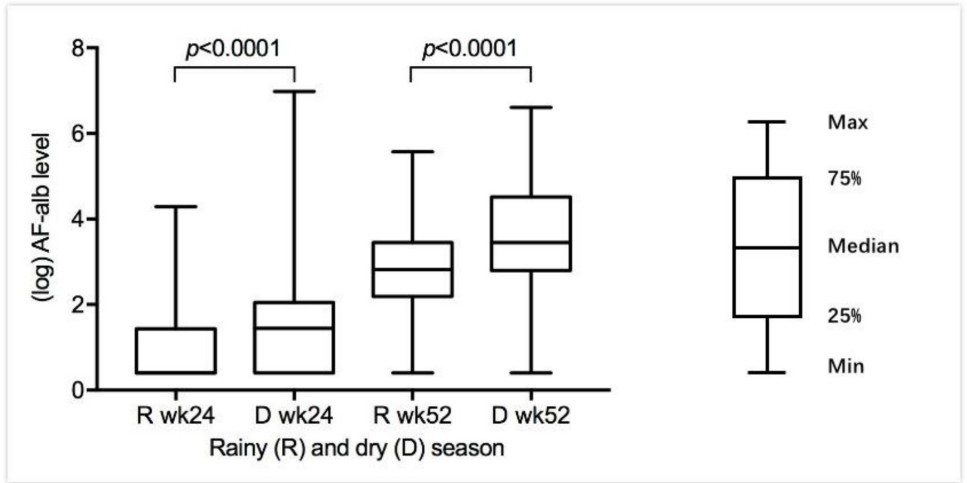

**Figure 1** Aflatoxin-albumin level (log transformed) in rainy and dry seasons at 24 weeks and 52 weeks. Wk, week.

vs 4.3 pg/mg, p<0.0001; 52 weeks: 19.3 pg/mg vs 35.5 pg/mg, p<0.0001).

### Thymic Index

Thymus growth showed an upward trend during the first 24 weeks of age (table 2). A significant increase was shown from birth to 8 weeks of age and reached maximum size at 24 weeks, followed by a decrease at 52 weeks. Seasonal variations were also identified in TI, with the TI values measured during the dry season being slightly higher than those measured in the rainy season at all time points, but this difference was only statistically significant in infants when measured at 8 weeks of age (mean: 14.3 cm$^3$ vs 13.3 cm$^3$, p=0.0136) (figure 2).

### Ab response to DTP vaccination

The Ab response to vaccination increased significantly at 24 weeks of age compared with 12 weeks (table 2). For pertussis, the protective Ab titres were detected in 69.3% of infants after the first dose, increasing to 96.0%

**Table 2** Mean level of TI and GM level of Ab response to vaccination in the ENID trial and the current substudy

|  | Current substudy | | ENID trial | |
|---|---|---|---|---|
|  | **n** | **GM (95% CI)/mean±SD/(%)\*** | **n** | **GM (95% CI)/ean±SD/(%)\*** |
| TI (cm$^3$) | | | | |
| Week 1 | 371 | 9.21±3.13 | 765 | 9.18±3.08 |
| Week 8 | 369 | 13.97±4.01 | 752 | 13.9±4.09 |
| Week 24 | 372 | 14.77±4.25 | 747 | 14.7±4.20 |
| Week 52 | 362 | 13.59±3.4 | 707 | 13.2±3.71 |
| Ab response to vaccination | | | | |
| Pertussis (EU/ml) | | | | |
| Week 12 | 355 | 7.61 (6.77, 8.57)/69.3% | 711 | 5.52 (5.04, 6.03)/50.1% |
| Week 24 | 322 | 132.37 (111.44, 157.22)/96.0% | 663 | 89.81 (77.71, 103.79)/88.2% |
| Diphtheria (IU/ml) | | | | |
| Week 12 | 355 | 0.26 (0.22, 0.32)/72.4 % | 711 | 0.12 (0.11, 0.14)/55.5% |
| Week 24 | 322 | 1.18 (1.04, 1.34)/94.1% | 663 | 1.40 (1.30, 1.51)/96.8% |
| Tetanus (IU/ml) | | | | |
| Week 12 | 355 | 0.83 (0.75, 0.91)/98.9% | 711 | 0.64 (0.59, 0.68)/97.3% |
| Week 24 | 322 | 4.34 (3.81, 4.94)/99.7% | 663 | 3.71 (3.40, 4.05)/99.6% |
| Morbidity | | | | |
| Sum of 12 weeks | 360 | 8.3±10.9 | 729 | 9.6±12.1 |
| Sum of 24 weeks | 361 | 18.7±18.3 | 730 | 22.8±23.5 |

International standards' protective Ab titres (WHO): diphtheria and tetanus >0.1 IU/mL[24 28]; pertussis >5.0 EU/ml.[27]
*The (%) presented that the percentage of samples had protective Ab titres.
Ab, antibody; ENID, Early Nutrition and Immune Development; GM, geometric mean; TI, Thymic Index.

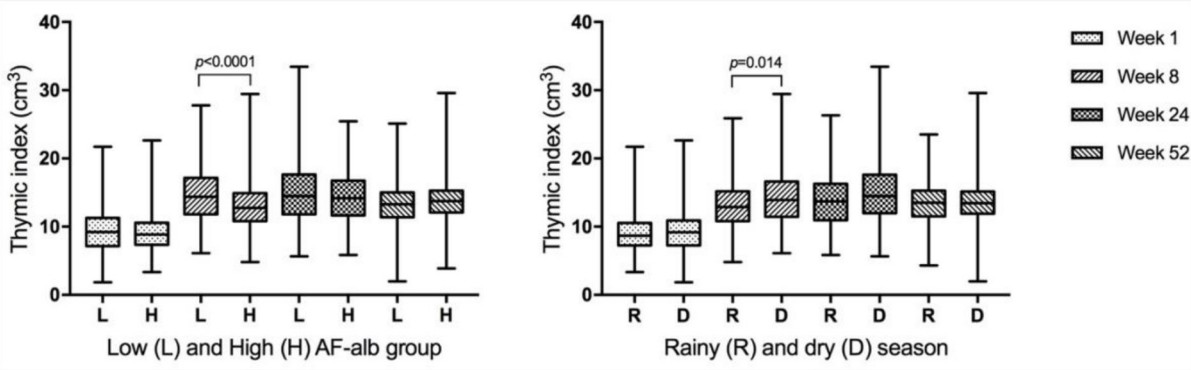

**Figure 2** Thymic Index (TI) in children in low or high aflatoxin-albumin (AF-alb) groups or in different seasons. The low and high AF-alb groups were divided by the median level at each time point. TI at week 8, week 12 and week 24 were grouped by AF-alb at week 24 and TI at 52 weeks was grouped by AF-alb level at week 52. The season of TI was defined as the time of measurement conducted.

at 24 weeks. For diphtheria, 72.4% of infants at 12 weeks and 94.1% at 24 weeks showed protective levels. For tetanus, the rates were 98.9% and 99.7% between the two time points. Ab titres were slightly higher in samples collected during the rainy season at 12 weeks of age, but lower at 24 weeks of age (figure 3).

### Primary outcomes of ENID trial versus ENID substudy

Table 2 compares the mean TI value and GM level of Ab titres in our AF substudy and the full ENID trial. Infants included in our substudy showed no significant difference in mean TI when compared with the overall 800 infants in the ENID trial. In terms of the Ab response titres, except for the Ab response to diphtheria at 24 weeks, both the GM of Ab titres and the percentage of samples with protective Ab titres were higher in the subgroup studied here than in the full cohort of ENID infants. However, morbidity was higher among the

ENID trial subjects than in our subsamples at both 12 weeks and 24 weeks.

### AF exposure and primary outcomes

Figures 2 and 3 present the TI and Ab titre in the low and high AF-alb groups. The mean TI was lower in the high AF-alb group than in the low AF-alb group during the first 24 weeks, but the difference was only statistically significant at 8 weeks (mean: 14.9 cm$^3$ vs 13.1 cm$^3$, p<0.0001), and not present at 52 weeks (figure 2). There were no significant differences for Ab response to pertussis and tetanus vaccines between the low and high AF-alb groups (figure 3). However, a significantly higher Ab titre against diphtheria was determined in infants in the high AF-alb group compared with those in the low AF-alb group at both time points (Ab GM at week 12: 0.16 IU/mL vs 0.46 IU/mL, p<0.0001; Ab GM at week 24: 0.96 IU/mL vs 1.46 IU/mL, p=0.0011).

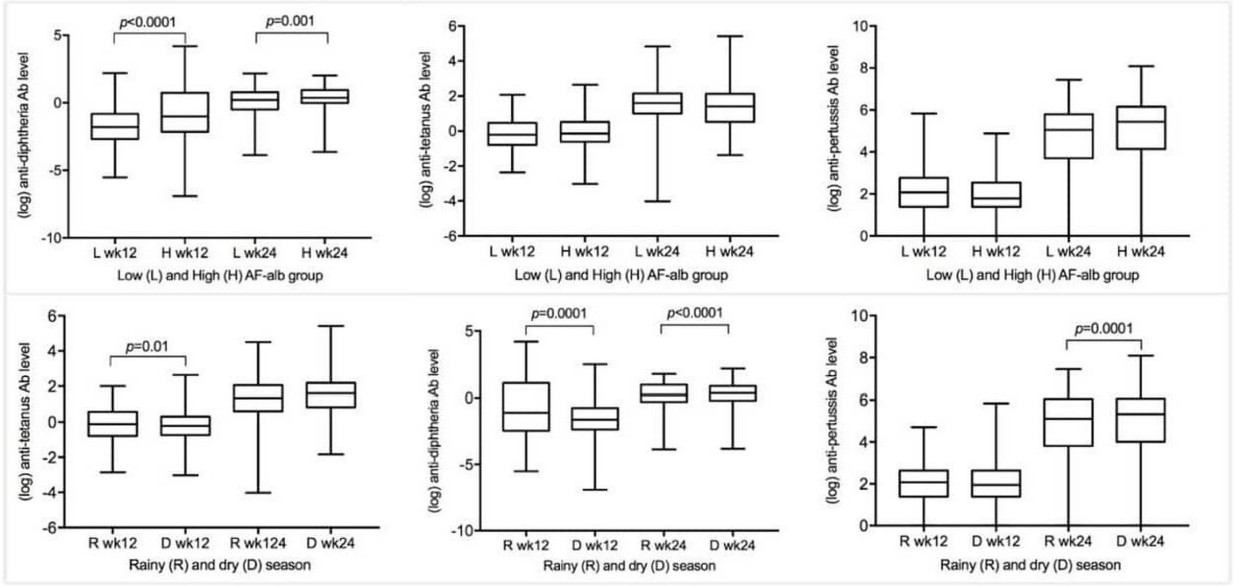

**Figure 3** Antibody (Ab) level to vaccination in low or high aflatoxin-albumin (AF-alb) group and in different seasons. The low and high AF-alb groups were divided by the median level at week 24. The season of Ab level was defined as the date of blood sample collection. Wk, week.

**Table 3** Relationship between AF-alb level and TI

| | Model 1 | | Model 2 | | Model 3 | |
|---|---|---|---|---|---|---|
| | Coef. (SE) | P value | Coef. (SE) | P value | Coef. (SE) | P value |
| Week 1 | −0.0083 (0.018) | 0.637 | −0.0012 (0.018) | 0.95 | −0.0012 (0.018) | 0.948 |
| Week 8 | −0.0606 (0.014) | <0.001 | −0.056 (0.014) | <0.001 | −0.056 (0.014) | <0.001 |
| Week 24 | −0.0098 (0.014) | 0.49 | −0.015 (0.015) | 0.3 | −0.015 (0.015) | 0.301 |
| Week 52 | −0.0115 (0.014) | 0.399 | −0.014 (0.014) | 0.296 | −0.014 (0.014) | 0.301 |
| All time points to infant age <24 weeks | −0.027 (0.011) | 0.014 | −0.023 (0.010) | 0.022 | −0.023 (0.010) | 0.021 |
| All time points to infant age <52 weeks | 0.0076 (0.007) | 0.28 | −0.053 (0.007) | <0.001 | −0.052 (0.007) | <0.001 |

TI at 1 week, 8 weeks and 24 weeks were analysed with AF-alb level at 24 weeks. TI at 52 weeks was analysed with AF-alb level at 52 weeks.
Model 1 unadjusted.
Model 2 adjusted for infant size (length), sex and season at TI measurement.
Model 3 adjusted for infant size (length), season at TI measurement, sex and maternal supplement groups (for age ≤24 weeks) and infants supplement groups (for age at 52 weeks).
AF-alb, aflatoxin-albumin; Coef., coefficient; TI, Thymic Index.

Regression analysis with or without adjusting for the other variables showed that there were no significant associations between AF-alb level and TI at 1 week, 24 weeks or 52 weeks (table 3). However, significant negative correlations were determined between AF-alb and TI at 8 weeks in all unadjusted and adjusted models. In addition, the random effect model, which measured the AF-alb correlation with all time points of TI pooled data, also showed significantly negative correlations at 24 weeks in both unadjusted and adjusted models, and at 52 weeks in adjusted models.

Regression analysis confirmed a significant positive correlation between AF exposure level and Ab response to diphtheria at both 12 weeks and 24 weeks of age (p<0.001 and p=0.002, respectively), but there were no significant associations between AF-alb level and Ab response to pertussis and tetanus (table 4).

Neither maternal nor infant supplementation had a significant effect on AF-alb level, TI or Ab titre at any time point (data not presented).

## DISCUSSION

This is the first study to investigate the impact of AF exposure on immune function of infants less than 12 months of age using TI and Ab response to DTP vaccination as

**Table 4** Relationship between AF-alb level and Ab response to vaccination

| | Model 1 | | Model 2 | | Model 3 | |
|---|---|---|---|---|---|---|
| | Coef. (SE) | p | Coef. (SE) | p | Coef. (SE) | p |
| **Pertussis** | | | | | | |
| Week 12 | −0.093 (0.058) | 0.11 | −0.093 (0.060) | 0.123 | −0.088 (0.070) | 0.207 |
| Week 24 | 0.143 (0.083) | 0.085 | 0.133 (0.086) | 0.124 | 0.099 (0.096) | 0.301 |
| Fold change | 0.190 (0.109) | 0.083 | 0.197 (0.113) | 0.084 | 0.127 (0.127) | 0.319 |
| **Diphtheria** | | | | | | |
| Week 12 | 0.421 (0.093) | <0.001 | 0.418 (0.094) | <0.001 | 0.495 (0.109) | <0.001 |
| Week 24 | 0.186 (0.058) | 0.002 | 0.190 (0.061) | 0.002 | 0.204 (0.064) | 0.002 |
| Fold change | −0.239 (0.097) | 0.014 | −0.124 (0.097) | 0.204 | −0.149 (0.106) | 0.162 |
| **Tetanus** | | | | | | |
| Week 12 | 0.065 (0.048) | 0.175 | 0.065 (0.049) | 0.182 | 0.096 (0.056) | 0.089 |
| Week 24 | −0.034 (0.062) | 0.578 | −0.064 (0.064) | 0.322 | −0.071 (0.070) | 0.316 |
| Fold change | −0.111 (0.081) | 0.172 | −0.096 (0.084) | 0.252 | −0.127 (0.092) | 0.168 |

Ab response to vaccination were analysed with AF-alb level at 24 weeks only.
Model 1 unadjusted.
Model 2 adjusted for WHZ and season of sample collection.
Model 3 adjusted for WHZ and season of sample collection, sex, maternal supplement group, Hb levels and morbidity.
Ab, antibody; AF-alb, aflatoxin-albumin; Coef., coefficient; Hb, haemoglobin; WHZ, weight for height z-score.

indicators of immune function. Infants who were exposed to higher levels of AF had smaller thymus during the first 24 weeks of age and had significantly higher protective Ab titres against diphtheria.

The observed high prevalence of AF exposure in the present study is consistent with previous findings in the same region of Gambia.[5 31 32] It is worth noting that the ELISA method used in this study gives AF-alb values that are approximately threefold higher than AF lysine values measured by LC-MS methods but that results between methods correlate for the same samples.[33] This issue was discussed by McMillan et al[11] and needs to be taken into consideration when comparing values obtained using the different methods. An increasing prevalence and upward trend of AF-alb levels with age were determined in our study. Infants at 52 weeks showed more than sevenfold higher levels of AF-alb than at 24 weeks of age. AF exposure could happen in utero through transplacental exposure, and the exposure level will increase during the first year when infants are gradually introduced to complementary foods.[2]

The Gambia has pronounced dry and rainy seasons, and previous studies have shown a seasonal impact on both growth and AF exposure.[6 34] In the present study, serum samples collected during the dry season had significantly higher levels of AF-alb than those samples collected during the rainy season. The annual dry season in the Gambia is a time of relative food availability, as food supplies from the previous harvest are usually plentiful. AF contamination in foods and crops tends to increase after a period of storage.[35] Populations are more likely to consume old grains towards the end of the dry (harvest) season, and foods which have been stored for a period are more likely to be contaminated. Our findings are consistent with previous studies in Benin and Guinea.[10 36] However, exposure can also depend on food type. In Senegal, Watson et al found that higher AF-alb levels in the harvest season compared with the postharvest season correlated with the recorded high consumption (4 days or more days a week) of contaminated groundnuts during the harvest period, with groundnuts being more susceptible to contamination than maize.[37]

Similar growth trend and seasonal variation of TI were previously reported in the Gambia by Collinson et al who also reported the consistently smaller thymus size in the rainy season, and significant difference at 8 weeks of age (p=0.001).[38] The significant retardation of thymus at 8 weeks found in children with high AF exposure might be due to it the time point that the thymus grows fastest. The sonographic method used in the current study to assess TI only represents an anatomical feature and not function, but previous studies in animals and infants have demonstrated a correlation between thymus size and lymphocyte proportion and function.[39 40] Therefore, smaller TI in infants could predict a lower immunity in the future. Previously, AF-induced damage on the thymus has only been investigated in animal models. A recent study reported that AF caused thymic histopathological lesions and pathological impairments in chickens, which had been fed with AF-contaminated feeds (34.3–134 µg $AFB_1$/kg corn feed) for 21 days and 42 days.[40] The reduction in thymus size and number of apoptotic lymphocytes occurred in a dose-dependent manner.

There was no previous study investigating the influence of AF exposure on Ab response to DTP vaccination. In our study, we found a significant positive association between AF biomarker level and Ab titre to diphtheria. An earlier study conducted in Gambian infants determined a similar weak but significant positive correlation between AF-alb level and one serotype of anti-pneumococcal Ab.[15] An animal study to induce anti-AF $B_1$ Ab in dairy cows, which combined $AFB_1$ with recombinant diphtheria toxin molecules injected into heifers, boosted the generation of anti-$AFB_1$ Abs.[41] The effect of AF on Ab response could vary depending on the feature of the vaccines, as early animal studies conducted in chicken and rabbits reported inverse influence of AF exposure level on Ab response to different vaccines.[42 43] The mechanism behind the significant positive correlation between AF-alb level and Ab titre to diphtheria needs further investigation.

One limitation of the current study was that the AF-alb levels were not measured at the birth of infants or in the maternal blood; so while the AF-alb biomarker integrates exposure over the previous 2–3 months,[44] AF-alb levels may not accurately reflect exposure in early weeks. However, Turner et al[5] reported a close association of AF-alb levels in maternal and infants' cord blood, and also reported a significant negative correlation between maternal AF biomarker level and child weight and height gain at the first year of life.[5] In our study, we found that the TI was significantly correlated with the weight of infants in the first year. Considering the AF-associated growth retardation determined in infants in the current cohort,[7] we suggest that AF could be a potential factor that influenced child growth and thymus development during the early stages of life.

A few previous studies have examined the effect of high levels of AF exposure on markers of immunity in human. Turner et al[15] reported a decrease in expression of salivary IgA in children from rural Gambia with detectable AF-alb levels compared with those with undetectable levels.[15] A lower percentage of T cells and B cells has been observed in participants with high concentrations of AF-alb in Ghanaians aged between 19 years old and 86 years old.[16 17] However, some animal studies of AF-induced effects on the immune system are inconsistent: Li et al[45] and Meissonnier et al[46] found no significant effect of $AFB_1$ on humoral immunity function in chickens and pigs.[45 46] While another study found increased expression of IgM and IgG in pigs dosed with high AF-contaminated feed.[47] The mechanism of the immunoglobulin rise is unclear.

In conclusion, we have demonstrated a significant effect of AF exposure on infant immune development, assessed by thymic size. Less consistent evidence was found for infants' Ab response to vaccination. This study

adds to evidence that AF exposure in infants can modify the immune response, and further efforts should be made to ameliorate dietary AF exposure across populations in SSA.

**Contributors** MR, YYG, AP and SM contributed to the original study design and implementation. YX, GC, PN and TF-J have contributed to laboratory analysis. YX wrote the first draft of the manuscript. All authors approved the final version of the manuscript.

**Funding** The research reported here was funded by the Meridian Institute (agreement number: 9678.0) and the Bill and Melinda Gates Foundation (grant number: OPP1066947). The Early Nutrition and Immune Development trial was supported by the UK Medical Research Council (MRC) (grant number: MC-A760-5Q×00) and the UK Department for International Development (DFID) under the MRC/DFID Concordat agreement. The dataset analysed in this paper is available from the corresponding author on reasonable request, and with appropriate additional ethical approvals, where necessary.

**Competing interests** None declared.

**Patient and public involvement** Patients and/or the public were not involved in the design, or conduct, or reporting, or dissemination plans of this research.

**Patient consent for publication** Not required.

**Ethics approval** Ethical approval for the ENID trial and the aflatoxin substudy was obtained from the joint Gambian Government/Medical Research Council Unit The Gambia ethics committee (SCC1126v2 and L2013.40).

**Provenance and peer review** Not commissioned; externally peer reviewed.

**Data availability statement** Data not included in the manuscript are available upon reasonable request.

**ORCID iD**
Michael Routledge http://orcid.org/0000-0001-9139-2182

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
