## [Reviewer comments · BMJ Open]

ARTICLE DETAILS

TITLE (PROVISIONAL)	Impact of dietary aflatoxin on immune development in Gambian infants: a cohort study
AUTHORS	Xu, Ya; Moore, Sophie; Chen, Gaoyun; Nshe, Patrick; Faye-Joof, Tisbeh; Prentice, Andrew; Gong, Yun Yun; Routledge, Michael

VERSION 1 – REVIEW

REVIEWER	Xue, Kathy The University of Georgia
REVIEW RETURNED	10-Mar-2021

GENERAL COMMENTS	The study examines the potential impact of aflatoxin exposure on development of immune system in Gambian infants. The general study design is fairly solid, with a good population size, as well as data available from a systematic clinical trial. However, likely due to the nature of the trial itself, the results from this study may be difficult to delineate from that of the original trial. There are also numerous concerns that may greatly impact the validity of the results. Detailed comments: Introduction: Pg 3 Ln 48-57: This paragraph does not seem to flow very well. Please re-organize the contents of this and the previous paragraph. Methods: Pg 4 Ln 21: According to information from Citation #21, the original study design planned for 1000 enrolled women. What may have caused the loss or exclusions in n? Pg 4 Ln 24: How were the 374 children selected? Was it based on any criteria, or was it randomized based on treatment groups? Please clarify. Pg 5 Ln 6-7: According to information from Citation #21, there are additional vaccinations scheduled on top of the DPT. Will these have any affect on the results? Please clarify. Pg 5 Ln 20-21: Subsequent text indicate that the first time point's AF-alb adduct is used as a variable for comparing the TI and antibody response of earlier time points. Pg 4 Ln 40 indicate that serum sample from 12, 24, and 52 weeks were used. Is there a
--

reason that the blood collected from week 12 is not analyzed for AF-alb adduct?

Pg 5 Ln 22-23: While ELISA method can be very efficient for quantification of AF-alb adducts, it has a likelihood of over-estimating the amounts due to the nature of antibody binding. A confirmation with LC-based method may be necessary to validate the results.

Pg 6 3-7: In lines 7-9, the author described the seasonal pattern that may potentially impact results, with approximately 3-4 months of rainy season, and the rest dry season. While it is a good idea to consider the impact of seasonal change on exposure and immune responses, it should note that there would be a high likelihood that the same child may have a flipped category in terms of the seasons on later measurements. Additionally, discrepancies in early life may impact the responses at later. Consequently, it may be likely that the rain group in one of the sampling points may not be the same on another, and may not be comparable.

Results:

Pg 7 Ln 33-36: The break-down of infant treatment-groups have been listed here, which is nice consider the potential impact of the treatment on the measured responses. However, that of maternal treatment was not. According to Pg 6 Ln 53-54, the mean duration of exclusive breast feeding was 5.2 months, equating to approximately 20 weeks, and the infant treatment regiment did not begin until 6th month. With measurements taken at weeks 1, 8, 12, 24, and 52, the first three before weaning, the maternal treatments need to be accounted for here.

Pg 7 Ln 45-48: Re-iterating the concerns regarding ELISA methods. The AF-alb concentrations are surprisingly high. Have the results been confirmed with LC-based methods?

Pg 10 Ln 43-45: The results in Citation # 26 indicate that supplementation does affect antibody levels. Please explain the discrepancy here.

Discussion:

Pg 12 Ln 27-30: Precipitation and humidity plays a role in impacting exposure. However, when it comes to immune responses, in terms of thyroid size, for example, there are multiple other causes that can result in significant differences. A very common example is that during the dry, winter seasons, people are more likely to get sick from flu. Has the author examined other potential causes for the differences in thyroid size and antibody counts, such as prevalence of commutable diseases?

Pg 13 Ln 10-14: Indeed, serum adduct level may be a proxy for exposures of 2-3 months ago. However, with the serum measurement taken for week 24 (approximately 6 months old), the measurement may not be representative for that of 3 months ago or earlier, i.e. that for weeks 1, 8, and possibly 12. Also, it should

	be noted that route of exposure for infants actually change drastically over short span of time, from in utero to nursing to weaning in a span of 5-6 months according to Pg 6 Ln 53-54, which is unlike typical adult exposures that consistently comes from diet.
--	---

REVIEWER	Sumarah, Mark London Research and Development Centre
REVIEW RETURNED	12-Apr-2021

GENERAL COMMENTS	The manuscript "Impact of dietary aflatoxin on immune development in Gambian infants" addresses the important topic of the role aflatoxin exposure on children. The final conclusions are the study are overly exciting but it is still an important dataset that should be published. The strength of the study is that it looks at a large number of subjects at multiple time points and is anchored into an even larger study. There are a number of weakness that should be addressed. The first is that the raw data for individual patients should be presented. The mean values based on the absolute values are high especially compared to studies that have used validated LC-MS/MS methods. Some discussion about the fact that ELISA is known to overestimate the true values vs. LC-MS would help address this. The reference list is too focused on the previous work from this group. There are many other studies that should be noted to better represent the state of the art in this field.
---

VERSION 1 – AUTHOR RESPONSE

Reviewer: 1

Dr. Kathy Xue, The University of Georgia

Comments to the Author:

The study examines the potential impact of aflatoxin exposure on development of immune system in Gambian infants. The general study design is fairly solid, with a good population size, as well as data available from a systematic clinical trial. However, likely due to the nature of the trial itself, the results from this study may be difficult to delineate from that of the original trial. There are also numerous concerns that may greatly impact the validity of the results.

Detailed comments:

Introduction:

Pg 3 Ln 48-57: This paragraph does not seem to flow very well. Please re-organize the contents of this and the previous paragraph. Elements of the introduction have been revised to address this.

Methods:

Pg 4 Ln 21: According to information from Citation #21, the original study design planned for 1000 enrolled women. What may have caused the loss or exclusions in n?

It may be that the reviewer is comparing the 374 subjects in this sub-study with the original ENID trial? Information on the original ENID trial is provided for completeness but the numbers are not relevant to the sub-study. This sub-study was powered to address the question of whether aflatoxin exposure was associated with differences in thymic index or antibody response. It involved blood samples taken from

children born into the ENID trial between May 2011 and December 2012 (N=374). The overall ENID study, addressed different questions and required a larger number of subjects.

Pg 4 Ln 24: How were the 374 children selected? Was it based on any criteria, or was it randomized based on treatment groups? Please clarify.

The 374 children recruited into the aflatoxin sub-study were all the children born into the main ENID trial between May 2011 and December 2012. There was no selection of these infants specifically other than the time period. We have added the word "all" into the methods (Study subjects, line 8) to clarify this. All children born into the ENID trial were randomised into treatment groups upon recruitment; those included in the current sub-study were also equally distributed to trial arms due to the sequential way they were recruited during the timeframe of the sub-study. We have added a sentence to Study subjects to clarify this.

Pg 5 Ln 6-7: According to information from Citation #21, there are additional vaccinations scheduled on top of the DPT. Will these have any affect on the results? Please clarify.

We have looked at the effects of the aflatoxin exposure on antibody response for DTP, which is being used as the marker of immune response in this part of the study. There is no reason to assume the other vaccinations will have affected results as these would be given to all children in the study.

Pg 5 Ln 20-21: Subsequent text indicate that the first time point's AF-alb adduct is used as a variable for comparing the TI and antibody response of earlier time points. Pg 4 Ln 40 indicate that serum sample from 12, 24, and 52 weeks were used. Is there a reason that the blood collected from week 12 is not analyzed for AF-alb adduct? At this very early stage the AF-alb levels in majority of serum samples are below the LOD, as the infants are not often given family food.

Pg 5 Ln 22-23: While ELISA method can be very efficient for quantification of AF-alb adducts, it has a likelihood of over-estimating the amounts due to the nature of antibody binding. A confirmation with LC-based method may be necessary to validate the results.

It is true that LC-MS or LC-FLD give lower values for AF-lys than the ELISA, which is reported as AF-alb. However, comparative studies have shown that the methods correlate with each other (McCoy et al, 2008, Cancer Epi Biom Prev 17, 1653-1657). The ELISA method used here has been validated against dietary intake of aflatoxin in more than one study (Wild et al 1992, Can Epi Biom Prev 1, 229-234; Routledge et al 2014, Biomarkers 19, 430-435) and has been used over many years for assessing health effects of aflatoxin exposure. So whilst LC-MS measures AF-lys very precisely, use of AF-alb as a biomarker of levels of aflatoxin exposure is validated and well established. We have added some additional comments in the discussion to reflect this.

Pg 6 3-7: In lines 7-9, the author described the seasonal pattern that may potentially impact results, with approximately 3-4 months of rainy season, and the rest dry season. While it is a good idea to consider the impact of seasonal change on exposure and immune responses, it should note that there would be a high likelihood that the same child may have a flipped category in terms of the seasons on later measurements. Additionally, discrepancies in early life may impact the responses at later. Consequently, it may be likely that the rain group in one of the sampling points may not be the same on another, and may not be comparable.

It is correct that exposure to one season at a particular age in infancy would automatically correlate to another time of year at a different age in infancy (or indeed fetal life). It is therefore not possible to determine whether any seasonal influences are due to season of assessment or another variable correlated with season of assessment (e.g. infants who are born in the wet season as low birth weight have spent much of their fetal life in the dry season, and it is likely that fetal growth restriction in the wet season months has contributed to the low birth weight in the dry season months). Within the current, observational study, seasonal trends are therefore discussed, but we cannot demonstrate causality, or pin point the timing of the effect. For this reason, we do not place major emphasis on season in this manuscript.

Results:

Pg 7 Ln 33-36: The break-down of infant treatment-groups have been listed here, which is nice consider the potential impact of the treatment on the measured responses. However, that of maternal treatment was not. According to Pg 6 Ln 53-54, the mean duration of exclusive breast feeding was 5.2 months, equating to approximately 20 weeks, and the infant treatment regiment did not begin until 6th month. With measurements taken at weeks 1, 8, 12, 24, and 52, the first three before weaning, the maternal treatments need to be accounted for here.

As detailed in the methods (Statistical analysis), the maternal supplementation was taken into account in the modelling. The breakdown of maternal supplementation (previously omitted) has now been listed in the table. This confirms the roughly equal distribution of infants between randomised supplementation groups (both infant and maternal).

Pg 7 Ln 45-48: Re-iterating the concerns regarding ELISA methods. The AF-alb concentrations are surprisingly high. Have the results been confirmed with LC-based methods?

In comparison to previous work the results are not surprisingly high, within the context that populations such as these have high exposure to aflatoxin in their diet. The results are consistent (within a range varying on local exposure) with previous studies in Gambia and other countries in sub-Saharan Africa (Xu et al 2018 World Mycotoxin Journal 11, 441-449). As discussed above, previous comparison of analyses by LC-MS, LC-FLD and ELISA have shown that AF-alb values obtained with ELISA are about 3 fold higher than AF-lys values obtained with LC-MS. However, results correlate across a range of biomarker values.

Pg 10 Ln 43-45: The results in Citation # 26 indicate that supplementation does affect antibody levels. Please explain the discrepancy here.

In the current sub-analysis, no effects were observed between supplement group (maternal or infant) and either the exposure variable (AF-alb) or outcomes (TI or Ab titre). In the full analysis of the ENID trial cohort, an effect of maternal supplement group was observed on antibody response to vaccination (Okala et al. 2019) and an effect of infant supplement group on TI (Moore et al, 2019). It is likely that these effects were not seen in the current analysis as there was insufficient power, based on the lower number of participants. However, examining the impact of the supplementation was not the purpose of the current sub-study.

Discussion:

Pg 12 Ln 27-30: Precipitation and humidity plays a role in impacting exposure. However, when it comes to immune responses, in terms of thyroid size, for example, there are multiple other causes that can result in significant differences. A very common example is that during the dry, winter seasons, people are more likely to get sick from flu. Has the author examined other potential causes for the differences in thyroid size and antibody counts, such as prevalence of commutable diseases?

The focus of this study was thymus size, not thyroid.

Seasonal effects on the outcomes were not the focus of the analysis but variables such as morbidity were taken into account in the modelling (detailed in the methods).

Pg 13 Ln 10-14: Indeed, serum adduct level may be a proxy for exposures of 2-3 months ago. However, with the serum measurement taken for week 24 (approximately 6 months old), the measurement may not be representative for that of 3 months ago or earlier, i.e. that for weeks 1, 8, and possibly 12. Also, it should be noted that route of exposure for infants actually change drastically over short span of time, from in utero to nursing to weaning in a span of 5-6 months according to Pg 6 Ln 53-54, which is unlike typical adult exposures that consistently comes from diet.

This is a good point and the text has been modified to acknowledge this limitation more clearly.

Reviewer: 2

Dr. Mark Sumarah, London Research and Development Centre

Comments to the Author:

The manuscript "Impact of dietary aflatoxin on immune development in Gambian infants" addresses the important topic of the role aflatoxin exposure on children. The final conclusions of the study are overly exciting but it is still an important dataset that should be published. The strength of the study is that it looks at a large number of subjects at multiple time points and is anchored into an even larger study. There are a number of weaknesses that should be addressed. The first is that the raw data for individual patients should be presented. The mean values based on the absolute values are high especially compared to studies that have used validated LC-MS/MS methods. Some discussion about the fact that ELISA is known to overestimate the true values vs. LC-MS would help address this. The reference list is too focused on the previous work from this group. There are many other studies that should be noted to better represent the state of the art in this field.

It is not normal practice to present raw data for individual subjects in a manuscript like this. The values are not high for AF-alb measured in populations such as this. Unfortunately, this reflects the high exposure to aflatoxin that is common in sub-Saharan Africa (Xu et al 2018 World Mycotoxin Journal 11, 441-449).

The reviewer makes a good point that we have been more focused on studies using the ELISA method to measure AF-alb in African populations and to address this have added two more references that have measured aflatoxin lysine by LC-MS in African children in recent years.

Previous comparison of analyses by LC-MS, LC-FLD and ELISA have shown that AF-alb values obtained with ELISA are about 3-fold higher than AF-lys values obtained with LC-MS. However, results between methods do correlate across a range of values (McCoy et al, 2008). If the study had been carried out using LC-MS then lower values would have been obtained (reported as AF-lys) but as the methods have been shown to give results that correlate (McCoy et al, 2008) and the ELISA has been validated against dietary intake of aflatoxin (Wild et al 1992, *Can Epi Biom Prev* 17, 1653-1657; Routledge et al 2014, *Biomarkers* 19, 430-435), this should not affect analyses based on the biomarker values obtained in this study. We have commented on the difference between LC-MS and ELISA in previous papers but have now cited the recent paper by McMillan et al to guide readers to a good recent analysis of this issue.

VERSION 2 – REVIEW

REVIEWER	Sumarah, Mark London Research and Development Centre
REVIEW RETURNED	27-May-2021
GENERAL COMMENTS	The authors have adequately addressed the reviews concerns. But I still feel that it is normal in a quality journal to present individual patient values from the ELISA in the supplementary data. This would allow readers of the paper to better understand the distribution levels and more importantly for reprocessing and interpretation of the data.